# Barriers to timely disclosure of HIV serostatus: A qualitative study at care and treatment centers in Dar es Salaam, Tanzania

**Neelam Ismail[1], Nancy Matillya[1] \*, Riaz Ratansi[1], Columba Mbekenga[2]**

**1** Department of Family Medicine, Aga Khan University, Dar es Salaam, Tanzania, **2** School of Nursing and Midwifery, Aga Khan University, Dar es Salaam, Tanzania

\* nancy.matillya@aku.edu

## Abstract

### Introduction

Disclosure of Human Immunodeficiency Virus (HIV) status is important to prevent the spread of HIV and maintain the health of people living with HIV, their spouses, and the community. Despite the benefits of disclosure, many people living with HIV delay disclosing their status to those close to them thereby increasing the risk for disease transmission. This study aimed to determine the barriers to timely disclosure of HIV serostatus for people living with HIV in Dar es Salaam, Tanzania, and identify what motivated disclosure.

### Methods

A qualitative descriptive study using in-depth individual interviews was conducted with10 participants attending HIV care and treatment centers in Dar es Salaam. The participants were people living with HIV who had delayed disclosing their serostatus for more than one month after diagnosis. Data was analyzed using qualitative content analysis.

### Results

Three categories emerged from the analysis: Barriers hindering timely disclosure, motivation for disclosure of serostatus, and consequences of delayed disclosure. Barriers to timely disclosure included denial of one's status, the fear of stigmatization, fear of being separated or divorced, the need to protect loved ones, and lack of adequate knowledge about the disease. Reasons that motivated disclosure included gaining social support, preventing disease transmission and wanting to be at peace.

### Conclusion

Timely disclosure is hindered by stigma because HIV is negatively perceived by the public. People living with HIV prefer not to disclose to avoid the negative consequences of disclosure, especially because of fear of being discriminated against and losing their social status, which plays a major role in social status in Tanzania.

Trust and adequate counseling from health care workers helps prompt disclosure.

**Data Availability Statement:** Data cannot be shared publicly because of the Aga Khan University research policy. Data is available from the Aga Khan University Institutional Data Access / Ethics

Committee (Associate Dean, Medical College Aga Khan University, Tanzania 2 Ufukoni Road, P. O. Box 38129, Dar es Salaam, Tanzania.) for researchers who meet the criteria for access to confidential data. The data underlying the results presented in the study are available from Ms Mwanaarab Sibuma available through mwanaarab.sibuma@aku.edu or +255682000972.

**Funding:** We received a USD 1000 student's grant from the Aga Khan University (www.aku.edu) for carrying out this study however the funders had no input into the design, data collection and analysis, decision to publish or preparation of the manuscripts. The authors had a right to publish regardless of the results. The views expressed are those of the author(s).

**Competing interests:** The authors have declared that no competing interests exist.

# Introduction

HIV is a burden for the healthcare systems worldwide and presents a major global public health issue that has taken the lives of more than 35.4 million people to date. Sub-Saharan Africa is the most affected region in Africa with 19.6 million people living with HIV/AIDS (PLWHA), which accounts for two-thirds of the global number of new HIV infections [1].

HIV is a major burden for Tanzania's health care system with an estimated prevalence of 4.5%. Data for 2017 showed that32,000 people died from an HIV related illness, 1.5 million people were living with HIV, and 65,000 people were newly infected with HIV [2].

HIV/AIDS has social, economic, and health impacts in Tanzania, a low-middle income country where the health care system has limited resources [3]. Prevention of HIV transmission is an important factor in reducing the disease, which can be achieved by disclosure of serostatus, particularly early disclosure.

Disclosure of HIV to a third person is defined as the process of a person revealing their HIV status, whether positive or negative [4].

Disclosure of one's HIV status to sexual partners is essential in limiting the transmission of HIV infection. It is also important in gaining social support from others and facilitating compliance and adherence to antiretroviral (ARV) medications [5]. However, disclosing HIV serostatus is not easy, as it is a personal and complex matter that is difficult to execute, especially as it is often associated with stigma [6]. Disclosure entails a process of communication about a stigmatized, life-threatening, and highly transmissible infection.

The ambitious 95-95-95 strategy was announced by UNAIDS in 2014, aiming to end the AIDS epidemic by 2030 by achieving 95% diagnosis among all people living with HIV (PLHIV), 95% on antiretroviral therapy (ART) among diagnosis, and 95% virally suppressed (VS) among treated. An intermediate goal of 90-90-90 was set for 2020 [7]. Disclosing serostatus means there is easier access and linkage to care and ARV medication, and therefore viral suppression can be achieved.

Women in Tanzania have a lower rate of disclosure compared with men [8]. This may be because they face stigma and discrimination following disclosure, which may be related to their weak social and economic status within their husband's family. They also fear violence because of gender inequality as disclosure can possibly endanger or destroy a relationship, leaving the women with an additional financial burden [6,9].

Delay in disclosure prevents PLWHA from reaping the benefits of disclosure and can result in further transmission the disease. In this study, delay in disclosure is defined as the time taken for disclosure of HIV serostatus being longer than one month since diagnosis of the disease. This definition was adopted as several studies found that most people diagnosed with HIV disclosed their status within the first month [6,8,10–12].

It has been reported that PLWHA weigh the risks (e.g. fear of abandonment and discrimination) against the benefits (e.g. need for support) of disclosure before deciding to disclose [4]. An individual's disclosure decision may also change over time, depending on motivating factors and the individual's situation and health which may lead to a delayed disclosure. In Ethiopia, a reported reason for delayed disclosure was that patients feared losing their social status and position that they had built for a long time. There was also the fear of discrimination by their relatives or sexual partners [13]. In South Africa, PLWHA delayed disclosure because they lacked skills for disclosure and felt they would not be able to handle negative reactions from the person they disclosed to [14].

In Tanzania, the rate of documented delay in disclosure to partners, family, and others was reported to range from 15–23% after one month of diagnosis [6,8] to partners, family and others.

A study in Dar es Salaam [15] found that there was a delay in disclosure before initiation of ARV therapy because people did not want to be stigmatized early in the disease as symptoms appear at a later stage. PLWHA also feared losing their social status in their social circle, which had been constructed over a long time. Another study conducted in Tanzania found that delay in disclosure among women was related to fear of violence because of gender inequality; disclosure of HIV to a husband could possibly endanger or destroy a relationship [6]. In Morogoro, Tanzania, it was found that a reason for delayed disclosure was that those who were diagnosed were in shock and were struggling with their HIV diagnosis and therefore felt unable to disclose in a timely manner [8].

However, although reasons for this delay have been studied using quantitative methods, there has been little evidence of in depth explorations. More qualitative information is needed to understand why PLWHA choose to delay their disclosure, and to identify reasons and methods used for eventual disclosure. Those who do not disclose in a timely manner forgo the numerous benefits of early disclosure, such as social support, prevention of disease transmission, ARV therapy adherence, viral suppression and early linkage to care. Previous studies from Tanzania highlighted the limited information on reasons why PLWHA delay disclosing their HIV status, and the need to further explore reasons hindering successful and timely disclosure, and to understand the motivation for eventual serostatus disclosure.

## Materials and methods

### Study design and participants

This qualitative descriptive study used individual in-depth interviews and content analysis [16] to explore barriers to timely disclosure of HIV serostatus. The theoretical framework used for this study was the Disclosure Decision Model (DDM): determining how and when individuals will self-disclose [17]. This framework was used to guide discussion related to barriers to timely disclosure and interpretation of the data. The DDM describes the disclosure process and what motivates and influences a person to disclose. It explores how the decision to disclose is based on an evaluation of the possible rewards versus the possible risks of disclosing in any specific social situation.

This study aimed to recruit participants from two care and treatment centers (CTCs) in Dar-es-Salaam: The Mnazi Mmoja Hospital and The Aga Khan Hospital. The Aga Khan Hospital is a private tertiary hospital that mostly receives patients mostly from the urban population in Dar es Salaam. In this hospital, provision of ARV medication is subsidized by the government and is provided free of charge; however, patients must pay for any tests (e.g., cluster of differentiation 4 (CD4) level, viral count) or consultations with the doctor. Physician-initiated testing and counseling is also charged to the patient.

Mnazi Mmoja Hospital is a public/government hospital where all CTC services (medication, tests, and counseling services) are supported by the government and provided free of charge.

Study participants included registered patients at both hospitals who were initiated on treatment, were aged 18 years and above, had a delay in disclosure of more than one month since diagnosis, and provided written informed consent. We excluded patients who were too sick to cooperate and answer questions as well as patients with debilitating mental illness.

Purposeful sampling was used to include participants who were considered most informed about the study topic. Participants from both sexes and different social/economic groups and education levels were included to ensure maximum variation and obtain broad insights and perspectives. Data collection continued until thematic saturation was achieved, which occurred after ten participants had been interviewed. All interviewed participants were from

Mnazi Mmoja hospital as no participants from The Aga Khan hospital consented to be part of this study.

## Ethical consideration

Scientific review of the proposal was sought from the Aga Khan University Research Committee (AKU-RC) and ethical approval from the Aga Khan University Ethics Review Committee (AKU-ERC).Permission to conduct the study at the hospital was obtained from the Medical Directors of both the Aga Khan Hospital and the Mnazi Mmoja Hospital.

Signed written informed consent for conducting the interview as well as being recorded was sought from the study participants before participation. The consent form was in English and translated into Kiswahili, the national language of Tanzania. The consent form included the standard details of confidentiality and protection of privacy.

## Data collection procedure

Participants who had disclosed their status after one month were identified during their monthly drug refill and approached by either the researcher or the CTC nurse. After an introduction about the study, interested patients that were willing to take part were enrolled using a recruitment checklist and then scheduled for an in-depth interview. All interviews were conducted in a counseling room at Mnazi Mmoja Hospital, with the door locked to prevent others entering and to maintain privacy. The in-depth individual interviews were conducted in Kiswahili language using an interview guide that explored participants' experiences of barriers and reasons for delaying disclosure of HIV serostatus(See supporting information S1 and S2 Files). The interviews took 30–40 minutes on average to complete and were recorded (with participants' permission) using an audio device which was transcribed verbatim for analysis. Data collection and analysis were undertaken iteratively, and any new, unexpected findings that emerged were incorporated into the process [16]. The interviews were initially conducted by both the main researcher (a final year family medicine resident) and a trained research assistant who was a native Kiswahili speaker with experience in conducting qualitative interviews. After two interviews and following discussion with supervisors, it was decided that the researcher should step back to allow more freedom for the interview discussion as participants were reluctant to talk openly because two people were present in the interview room.

Participants were informed they could decline to participate in this study without prejudice; they were assured that refusing to participate would not influence their access to services when attending regular CTC appointments for check-up and drug refills. Those who participated were given 10,000 TSH (approximately USD 4.3) after the interview as a gesture of appreciation for their time. Field notes on general impressions, contextual matters and non-verbal communication were taken. At the end of each interview, the interviewees were invited to ask questions about the study. All in-depth interviews were anonymized by allocating each participant a number, which was then used for all subsequent analyses and reporting the results.

## Data analysis

Data analysis was undertaken out using qualitative content analysis [16]. Interim data analysis occurred concurrently with data collection, with supervisors providing regular feedback to the main researcher. Emerging issues and further data collection needs were identified and incorporated into data collection. All interviews were transcribed and translated into English by a native Kiswahili-speaking nurse who had prior experience in translating and transcribing qualitative research. The transcripts were checked for accuracy against audio recordings to detect any mistakes and changes were made as necessary.

**Table 1. Example of the analysis process.**

| Meaning unit | Condensed meaning unit | Codes | Subcategory |
|---|---|---|---|
| "I was so afraid of how he (husband) would receive the information. After all my marriage was still young, just one year, yeah, so I was very afraid that my marriage would end that is why I hesitated to tell him for over a month" | Delayed disclosing because of fear of marriage breaking given that their marriage was 1 year old and was afraid of husband's reaction | Delay in disclosure because of fear of destroying marriage. Worried about husband's reaction | Fear of being divorced or losing a partner * |

* This subcategory was later included in the category "Barriers hindering timely disclosure".

The analysis comprised of repeated readings of the individual interviews to obtain a sense of the whole and to identify meaning units. Next, the identified meaning units were condensed into short, summarized versions (condensed meaning units). Thereafter, codes were formed, which were then grouped to form sub-categories. Finally, from these subcategories and reading back and forth, categories emerged that encompassed the subcategories and reflected the latent content of the text. The whole analysis process was conducted by the main researcher with input from the three supervisors, one of whom was an experienced qualitative researcher. This ensured that data analysis and emerging findings were grounded in the data. An example of the analytic process is shown in Table 1.

## Checking for trustworthiness

Four criteria were used to check for trustworthiness: credibility (internal validity), transferability (external validity), dependability (reliability) and, conformability (objectivity) [16,18].

Credibility was assured by creating an environment to promote openness and candid discussions with participants because HIV is a sensitive topic. Triangulation was achieved by using both field notes and transcribed data, engagement with the participants by clarifying points and using probing interview questions and checking the transcriptions. The use of direct quotes from participants to support the text description added to the credibility of the findings. Transferability and dependability were achieved by describing the methods in sufficient detail to create an audit trail, and providing clear and detailed descriptions of the selection and characteristics of participants, data collection, and analysis process. Conformability was ensured by triangulation and reflexivity, which was achieved by regular discussions with the supervisors throughout the research and analysis process. Regular debriefing meetings between the interviewer and the main researcher allowed for constant reflection during the data collection process, ensuring the study objectives were met.

## Results

Ten participants completed individual interviews. Participants' ages ranged from 26 to 45 years, and there were six females and four males. Their education ranged from primary school to college/university degree. The time taken from HIV diagnosis to disclosure ranged from 1 month to 8 years. All participants had disclosed to a family member, except for two participants who disclosed to their boss and a friend. All of the participants had one sexual partner and had no prior HIV status knowledge at time of disclosure.

Participants' demographics on disclosure are presented in Table 2 below.

## Categories

Three categories emerged from the data:

1. Barriers hindering timely disclosure.

**Table 2. Participants' demographics and disclosure details.**

| Participant Identification number | Sex | Age | Marital Status | Economic Status | Level of education | Time to disclose | Disclosed to |
|---|---|---|---|---|---|---|---|
| 1 | F | 43 | Divorced | small business | Primary | 1 year | Husband/sister |
| 2 | F | 34 | Married | Unemployed | Secondary school | 8 years | Mother/niece |
| 3 | F | 37 | Married | Banker | Degree | 1 month | Husband |
| 4 | F | 37 | Married | Petty business | Secondary school | 3.5 months | Mother, siblings. and husband |
| 5 | F | 40 | Co-habiting | Social worker | Degree | 3 months | Current partner |
| 6 | M | 42 | Married | Unemployed | Diploma in BA | 3 months | Wife |
| 7 | M | 44 | Married | Petty business | Primary school | 2 months | Friend |
| 8 | F | 26 | Married | Petty business | Secondary school | 1 month | Mother |
| 9 | M | 28 | Single | Unemployed | Secondary school | 2 months | Uncle |
| 10 | M | 45 | Divorced | Employed | Primary school | 4 years | Boss |

2. Motivation for disclosure of serostatus.

3. Consequences of delayed disclosure.

**Barriers hindering timely disclosure.** Once diagnosed and informed of their results, a majority of participants were in denial and experienced an array of emotions including shock, surprise, feelings of loneliness, and sadness. Participants were unable to accept their status, which led them to delay disclosing it to others.

"*I was stunned! I did not expect those results. I was not sick, I was not ill, it was just general body malaise...I was just dumbfounded...*" P4

"*I did not expect that I would get the results like that...But I got the positive results, but I didn't really accept that situation and that was the reason I did not disclose this to anyone in my family for a long time.*" P 1

A majority of them also noted that they were not equipped with adequate knowledge about the disease and did not know how to disclose it to anyone. Many of the participants expressed feeling lonely and isolated as they lacked the skills to disclose.

"*We are not very well educated about the matter.*" P4

A major barrier to disclosure of HIV status was fear of the stigma associated with and experienced by those with the illness. They feared being outed, pointed at, and belittled by those they disclosed to.

"*The issue there is being outed. Most people including myself are afraid of being known, laughed or belittled and finger-pointing.*" P 2

They also feared being left or divorced, which made them not want to disclose to their sexual partner.

"*I was very afraid that my marriage would end that is why I hesitated to tell him for over a month.*" P 3

"*The real challenge was to the father of my children. How would I begin? I worried a lot about how he would receive the news...*" P 8

Participants were selfless. They were hesitant in disclosing their status to those close to them, because they were afraid of how they would react and wanted to protect them from heartache. They were also scared of losing their trust.

"*I was so afraid of telling my mother because I knew that if she found out she would feel very bad and would get hypertension from the news.*" P 2

There were some undesirable reactions after disclosure experienced by the participants that led to negative outcomes such as loss of support, attempted suicide, being rejected from their partner by being left/divorced, being blamed, financial troubles, and facing discrimination. Discrimination experienced by participants included lack of confidentiality, being belittled, and exclusion from social gatherings. All these factors acted as barriers to trust and to further disclose to others.

"*I had my partner before. I disclosed to him and this is the reason that made me not to disclose early, because as soon as I told him he divorced me. . . There are a lot of challenges, considering I have kids so sometimes I get a lot of financial constraints. . . My ex-husband does not help me with their upbringing and school fees.*" P1

"*I told my niece who took it positively but my niece eventually spread the word about my status. . .My niece told everyone in the family especially ones that didn't know. . . Family can really hurt with words and everyone will talk about you and laugh at you constantly.*" P2

"*My partner blamed me. . .After that, he stopped communicating with me. You could tell that he was just concerned about the child and had no affection for me and that is when it ended there.*" P5

One participant experienced severe discrimination that led to attempted suicide. With no support by his side and facing discrimination, he turned to living on the streets and finally attempting suicide because he had feelings of worthlessness.

"*My uncle told his wife. . . first they stigmatized me. Severely. They regarded me as someone who was going to die. There was no support. The hatred went all the way to his mother as well. . .it affected me; I ended up leaving home and living on the streets. They do not see my benefit, I feel unworthy. Ever since I have lived in the streets, not even one person has come looking for me. Education wise, I have already given up, I ended up in form four. So, for now no more school for me. I did not do exams because I had school debt. After he found out about my health status he stopped (paying). I attempted suicide. I tried to hang myself, fortunately, God did not want me to die. Somebody saved me. I was tired of feeling worthless, why should I continue suffering? The mind starts running games on you. I have no mother, no father; I really wanted to make up my mind. I left the front porch and went at the back of the house where it was very dark, it was around 8pm. I found a strong shawl, made a noose on my neck, but things did not turn out the way I wanted them to. I shouted, they came, found me and took me to the hospital.*" P 9

**Motivation for disclosure of serostatus.** Participants were asked about what made them eventually disclose after a long period of keeping their status to themselves. Responses varied depending on participants' situations and circumstances.

Some participants were seeking support in terms of financial aid, emotional support or medical support from those whom they disclosed to. They wanted the person they disclosed to

help them if they were to fall sick, and someone to be their spokesperson during their illness. That person would also be a source of emotional support and be able to collect medication on their behalf.

*"The main reason of disclosure of my HIV status to my sister is that there is a possibility of me falling sick and I will need her as a companion for support for getting my medications. . .even when I am sick she will know how and what to tell other relatives."* P1

Participants were in search of peace of mind and freedom from the need for secrecy in taking pills and attending clinic visits. Many expressed the fear of losing the trust of their partners if their status was revealed from other sources.

*"I felt like I had to tell him because, firstly, I wanted to be at peace because I was on medication. . . We live in one house; one room and I am using ARVs in secret. What if he found out accidentally one day and saw the medication?"* P3

Despite being scared, participants felt the need to inform their partners out of love for them by encouraging them to go and get tested to protect them from the disease and stop the transmission of the disease.

*"I used to get lost in thought, what if he is not infected, and we kept on having sex? That is when I decided that I had to tell him, in case he was not infected, then I might save him. . ."* P3

Other participants waited to disclose their status until their health had deteriorated, and they could no longer keep their condition a secret. When the signs and symptoms started to show, questions would be raised and eventually circumstance meant they had to disclose.

*". . . I kept it a secret for a long time until when the symptoms showed. . .during the whole time, I never got ill. Didn't show symptoms or anything. But in 2010 I started losing weight drastically and it showed. The fevers and malaria became very frequent. I had to disclose then."* P2

Participants noted that counseling from healthcare workers prompted them to disclose and added that receiving proper counseling about their condition and relevant education helped in disclosure and treatment support. Participants tended to disclose to people close to them, who they trusted, such as a family member, partner, or close friend.

*"The reason that made me open up to her is that when we came here, we were counseled well and tested."* P2

*"I felt at ease telling my siblings because I consider them my confidants"* P4

*"She (my wife) is my significant other. We have to stand together; you cannot tell anyone else except her. You can tell even your sibling and they would not get it as your partner- it was in me to confide in someone I trust."* P6

After disclosure participants experienced a myriad of reactions from their trusted ones, both positive and negative. Having experienced a positive reaction after disclosure resulted in support, and participants mentioned that compliance to treatment and managing symptoms was easier with that positive support. It also encouraged participants to disclose to others.

"*My mom insists on telling me daily that I should be strong and not feel sorry for myself. . .she follows up my refills and encourages me to take the medication.*" P2

"*Everyone received the information with a heavy heart truthfully. . .The most important thing is they encouraged me, telling me examples of how other people were surviving and that it is not the end of my life. They really walked with me in every aspect through it all. They would also remind me to attend the clinic. I was also, very open to them about everything. After I confided in my siblings, they advised that I should also tell my husband so we can use medication together.*" P4

"*I told my friend after two months. . . I felt very relieved. I felt free. I told my very close confidant. He was proud of me. He made me stronger by also assuring me about the importance of telling my wife. You see? It was not easy telling my wife. It was easy telling my friend, we grew up together.*" P7

**Consequences of delayed disclosure.**   Participants shared some negative consequences that they had experienced because of their lack of timely disclosure. Before disclosing, participants had engaged in regular unprotected sexual intercourse with their regular partner as they could not enforce condom use. This might have led to an increased risk of the transmission of the disease.

"*I told him at night in bed. You know being married and all, we must make love without condoms.*" P3

It was also noted that there was poor compliance with treatment and adherence to medication before disclosure because of the lack of support from those around them and the need to hide medications. This overall secrecy was hard for participants to maintain, and affected their relationship with their partner and cause distrust.

". . . *I did not even care about taking medication or disclose to anyone, I had no support.*" P10

"*I grew restless not sharing my status with him (partner) even when I leave him home alone I always wondered and worried if he could be looking around and scavenging my bags or handbags where I hide my things. So, when I knew he was home I would always come back very fast and just to check what was going on.*" P 5

## Discussion

All participants revealed that disclosing HIV test results was not an easy decision; rather it was a complex and difficult personal matter that entailed communication about a potential life threatening, stigmatized and transmissible infection. The findings of this study demonstrated that the decision to disclose changed over time, depending on the individual's situation and circumstances. According to the DDM [17], the first step in disclosure is entering the situation in which a disclosure goal is made salient or accessible; these goals motivate one to disclose. This study suggested that the goals for disclosure were gaining care and support from a confidant during illness, being considerate of others by preventing the transmission of disease, seeking peace of mind and freedom from secrecy, and participants' health deterioration. These considerations resulted in the disclosure of serostatus. Consistent with previous studies, these findings highlighted that the goals of disclosure included limiting transmission of the disease, gaining emotional and financial support, gaining support in adhering to medication, and being able to use medication freely [5,6].

Adherence to ARVs is important to achieve a suppressed viral load, which interrupts onward HIV transmission to susceptible partners. A previous study conducted in Dar es Salaam revealed that the disclosure of HIV status before initiation of ARV therapy improved patients' adherence and had a positive influence on CD4+ T-cell counts recovery as well as viral load suppression [15].

It has also been noted that disclosure of serostatus depends on the individual's state of health, with HIV-infected individuals delaying disclosure until their disease had progressed and it became difficult to conceal their illness from their partners [8,19].

External factors can also influence social goals around disclosure. This study found that participants disclosed their status to others, albeit delayed, if they were prompted and appropriately counseled by CTC nurses. This professional support helped to empower them with knowledge about their condition. This finding was consistent with a study from Ethiopia that demonstrated strong counseling services prompted disclosure [20]. In contrast, participants who were not counseled properly or had inadequate knowledge about their condition tended to delay disclosure. This finding was similar to a study conducted in South Africa that reported disclosure was delayed because of a lack of disclosure skills and support from health care workers to prepare patients for disclosure [14].

This is an important issue as it demonstrates that when the PLWHA do not receive adequate support and counseling after learning about their status they are less likely to disclose. The support and counseling they receive from health care providers empowers them to accept their serostatus, disclose this status to those they trust, and access benefits from early disclosure. Strengthening counseling services provided by health care providers by equipping them with adequate knowledge and counseling training can help timely disclosure among PLWHA.

The second stage of the DDM involves decisions about whether disclosure is an appropriate strategy to exercise, and with whom. The desire to avoid or delay disclosure of HIV status is influenced by the relationship with the target person. In this study, it was shown that participants would disclose this sensitive information to confidants that they relied on, trusted, and were close to. This finding was consistent with other studies from the US [10,21] that noted participants selected targets they trusted and were close to.

The next stage of disclosure is assessing the subjective utility of disclosure versus the subjective risk from disclosure. Subjective utility refers to the perceived value of the desired outcome to the individual who is disclosing, which was a similar finding in this study and previous studies [22]. Participants sought positive outcomes such as the need to be accepted by confidants and treated with respect as well as wanting to receive support after their disclosure.

The subjective risk from disclosure in this study was the perceived anticipated possible risks and negative outcomes encountered after disclosure, which formed the barriers to disclosure.

The biggest barrier faced by participants in this study was fear of stigmatization. They were worried about and fearful of social rejection and stigmatization by their confidants and families following disclosure, which would lead to loss of social support. Stigmatization was perceived in many forms, including social stigma, being laughed at or labeled, being left/divorced or being discriminated against. There also was a component of fear about being stigmatized early in the disease because the medication masked the physical effects of HIV progression.

The majority of participants in this study found out their status during their routine antenatal care when pregnant and others when they fell sick. Tanzania incorporates HIV testing and counselling in antenatal and reproductive and child health services and recommends engagement with, testing of, and counselling partners at health facilities [23]. For all participants it was the testing for the first time and so did not have prior knowledge on their status or when and where from they contracted the disease. On disclosure it lead to disagreements, loss of trust, strain in the relationship between partners and in some cases a breakdown of the relationship.

Similarly, several previous studies demonstrated that there were risks in disclosure, such as fear of stigma and abuse, fear of conflict, and fear of breach of confidentiality (i.e. betrayal) [8,15,24–26]. Other studies showed that delayed disclosure was attributable to fear of early stigmatization before the symptoms appeared because of the effectiveness of ARVs in concealing symptoms [13,27].

A striking finding was that the majority of the participants faced stigmatization (e.g. being laughed at, being discriminated), experienced infidelity, and one participant had attempted suicide.

This fear of stigmatization and the finger-pointing associated with HIV meant that all participants were hesitant in disclosing their serostatus. This highlighted that stigma was a major barrier to the disclosure process. PLWHA are faced with discrimination and loss of the social status that they worked hard to build within their society throughout their lifetime.

In a country such as Tanzania, social relationships are highly valued. The use of relationships to obtain benefits and achieve desired ends has been termed "social capital." The components of social capital are trust, cooperation, reciprocity, and sociability [28]. Stigma is feared because it leads to social isolation, thereby undermining relationships that are essential for survival. Avoiding HIV-related stigma therefore can be understood as an effort to conserve social capital.

Other studies in Tanzania have also demonstrated this aspect. After disclosure, individuals face stigma, discrimination, fear of their partner's reaction, and fear of a fall in social status, facing oppression, and divorce [8,9].Women are particularly vulnerable in this regard because of their weak social and economic status within their husband's families.

Another major finding from this study was that disclosure occurred after participants' acceptance of their own status. Participants had to mentally come to terms with their serostatus before they could disclose it to anyone. Their medication adherence also improved after acceptance. Similar findings were reported in other studies where disclosure took place after an individual's serostatus acceptance [8,21,29].

Health care workers play a prominent part in counseling PLWHA, which integrates acceptance of serostatus and equipping them with knowledge about their disease. This prompts self-acceptance leading to eventual disclosure.

A limitation of this study was that it became a single-center study despite initially (at the proposal stage) intending to include two hospitals (The Aga Khan Hospital and Mnazi Mmoja Hospital). However, individuals approached at The Aga Khan Hospital were reluctant to participate, stating reasons such as not having time, not wanting to be seen, and being wary of confidentiality.

There were several possible explanations for these responses. The Aga Khan Hospital caters to a higher social class of patients and this status of clientele may have an increased level of stigmatization around them. It was also thought that counseling provided by the hospital CTC might not have been adequate regarding the stigma of HIV, which could have made patients less receptive to discussing their experiences with HIV and the disclosure process.

A previous systematic review that discussed the role of social class on stigma reported that although it is recurrently implicated in HIV-related stigmatization, social class does not receive much notice in literature and is a neglected area of research [30]. This suggests that more studies are needed to specifically address stigma, particularly in private hospitals that cater for patients from higher social classes.

Another limitation of this study was the sampling method, which might have introduced selection and recall bias, especially for those who had a long period since they disclosed.

## Conclusion

Although it has been 30 years since HIV was first discovered in Tanzania, PLWHA still feel negative after effects of the stigma associated with the disease.

Timely disclosure is essential in minimizing the risk and preventing further transmission of HIV via sexual transmission to partners, as well as to improve ARV therapy adherence to support viral suppression. These factors would help achieve two parameters of the 95-95-95 target set by UNAIDS; namely, easier access and linkage to medications and subsequent viral suppression by 2030.

Timely disclosure is hindered by stigma. Stigma was the greatest barrier to disclosure identified in this study. HIV is negatively perceived by the public and PLWHA prefer not to disclose to avoid negative aspects such as being outed, risking a breach in confidentiality, being labeled, discriminated against, and loss of social status, which has a prominent role in Tanzania. Given that HIV is a highly stigmatized epidemic with multiple layers (gender, social class, sexual orientation, race, hyper sexuality), further qualitative studies are needed to help understand the role of stigma in social class among PLWHA in Tanzania and how to identify strategies to reduce stigma.

Given that adequate, efficient, and supportive counseling leads to self-acceptance, empowerment of PLWHA, and timely disclosure, more efforts need to be directed to ensuring quality counseling services at CTCs. This can be achieved by auditing and assessing the counseling provided by health care workers. Additional support should also be provided in terms of training for health care workers.

## Supporting information

**S1 File. Interview guide: English.**
(DOCX)

**S2 File. Interview guide: Kiswahili.**
(DOCX)

## Acknowledgments

We would like to thank the participants for their support in taking part in this study and sharing their insights into their experiences of the delay in disclosure. We would also like to thank and acknowledge the two health facilities (The Aga Khan Hospital and Mnazi Mmoja Hospital) where this study was conducted and extend our gratitude to our interviewer and research assistant.

## Author Contributions

**Conceptualization:** Neelam Ismail, Nancy Matillya, Riaz Ratansi, Columba Mbekenga.

**Data curation:** Neelam Ismail.

**Formal analysis:** Neelam Ismail, Columba Mbekenga.

**Funding acquisition:** Neelam Ismail.

**Investigation:** Neelam Ismail.

**Methodology:** Neelam Ismail, Riaz Ratansi, Columba Mbekenga.

**Project administration:** Neelam Ismail, Nancy Matillya, Riaz Ratansi, Columba Mbekenga.

**Supervision:** Nancy Matillya, Riaz Ratansi, Columba Mbekenga.

**Visualization:** Neelam Ismail, Nancy Matillya.

**Writing – original draft:** Neelam Ismail, Nancy Matillya, Riaz Ratansi, Columba Mbekenga.

**Writing – review & editing:** Neelam Ismail, Nancy Matillya, Riaz Ratansi, Columba Mbekenga.

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
