## [Decision Letter · Decision Letter 0]

19 Jan 2021

PONE-D-20-37341

Barriers to timely disclosure of HIV serostatus: A qualitative study at care and treatment centers in Dar es Salaam, Tanzania.

PLOS ONE

Dear Dr. Matillya,

Thank you for submitting your manuscript to PLOS ONE. After careful consideration, we feel that it has merit but does not fully meet PLOS ONE’s publication criteria as it currently stands. Therefore, we invite you to submit a revised version of the manuscript that addresses the points raised during the review process.

We look forward to receiving your revised manuscript.

Kind regards,

Joel Msafiri Francis, MD, MS, PhD

Academic Editor

PLOS ONE

Journal Requirements:

2. Please include a copy of the interview guide used in the study, in both the original language and English, as Supporting Information, or include a citation if it has been published previously.

3.We note that you have indicated that data from this study are available upon request. PLOS only allows data to be available upon request if there are legal or ethical restrictions on sharing data publicly. For information on unacceptable data access restrictions, please see http://journals.plos.org/plosone/s/data-availability#loc-unacceptable-data-access-restrictions.

Additional Editor Comments:

The reviewers provided positive recommendations and some critical issues to be addressed especially those related to the writing style of the paper.  

Reviewers' comments:

Reviewer's Responses to Questions

**Comments to the Author**

1. Is the manuscript technically sound, and do the data support the conclusions?

Reviewer #1: No

Reviewer #2: Yes

2. Has the statistical analysis been performed appropriately and rigorously? 

Reviewer #1: Yes

Reviewer #2: N/A

3. Have the authors made all data underlying the findings in their manuscript fully available?

Reviewer #1: No

Reviewer #2: No

4. Is the manuscript presented in an intelligible fashion and written in standard English?

Reviewer #1: No

Reviewer #2: No

5. Review Comments to the Author

Reviewer #1: This qualitative study is an exploration of the barriers and motivation for disclosure of HIV status to other by people living with HIV in Tanzania. This topic has the potential to strengthen HIV prevention programs and to inform programs that can facilitate effective HIV disclosure to others. Nonetheless, the paper needs more work to be suitable for publication. Below are areas that authors need to address to strengthen the paper.

Introduction

1. Line 59-61: Authors need to make it clear that disclosure of HIV status is one of the strategies that can help to prevent the transmission of HIV. The way it has been written, it is like disclosure of HIV status is the only way that can help to prevent the transmission of HIV.

2. The sentence covered by lines 65-67 needs citation.

3. It is important to describe the operational definition of HIV disclosure in this study as this can help to guide the interpretation of the findings.

4. The justification for conducting this study is not convincing or correct. Authors identify lack of studies that have assessed barriers to timely disclosure. However, contrary to the authors’ claim there are several studies in Tanzania that have been conducted related to barriers to disclosure of HIV status to others or sexual partners some of them include the following (Damian et al., 2019; Hallberg et al., 2019; Maluka, 2014). Thus, authors need to be clear with the reasons for conducting this study. Why is it important to conduct this study? What will it add to the existing literature regarding disclosure of HIV status to others/sexual partners in Tanzania?

Materials and methods

5. Where did data collection take place in the research site? How were participants approached to take part in the interview? How long did each interview take to be completed? How many people were involved in the data analysis process?

6. Given that no interview took place at Aga Khan Hospital, it is not necessary for the authors to include information about this facility in this article.

7. What type of questions were included in the interview guide. It would be important if authors can include the interview guide as supplementary material to their submission

Results

8. The findings in the three categories seem to be limited. For example, the last category ‘Consequences of delayed disclosure’ is too brief and lacks sufficient detail to be a standalone category. As a qualitative study, the findings of the study should have some depth to enable the readers to understand participants’ perspectives. The findings are also missing some important information that might be relevant to the study aim for example, what prompted participants to disclose their HIV status to the people they disclosed to? How did the significant others/family members reacted to the disclosure? How was the disclosure done?

Discussion

9. Authors have discussed some findings which are not presented in the findings section. For example, line 316-318, the authors have indicated that a majority of the participants experienced some sort of stigmatization (getting laughed at, discrimination); infidelity and one participant attempted suicide. This information is not described in the findings section.

10. The limitations of the study need more work. Aga Khan Hospital is not part of the research site in this study as none of the participants was recruited from this research site. Researchers should highlight the limitations of the research methodology used in this study.

11. The implications of findings for practice, research and policy are also not clearly explained in this section.

12. Lines 352 to 256, authors have discussed social class in relation to stigma. However, social class has not been presented in the findings of the study. Authors need to discuss or make research recommendations based on the findings of the study.

General comment

13. All sections of the paper need editing.

Reviewer #2: Congratulations on an interesting study, the findings of which should be of interest to others working in Africa.

The methods you have used and have described in your paper are sound and appropriate.

The results, whilst not new, are important as they highlight the need for consistent high quality counselling of PLWHA in order to disclose early and interrupt onward transmission to sexual partners.

The points you raise in your discussion are clear and reference other appropriate and relevant research. I suggest you consider including in the discussion some thoughts about the importance of adherence to ARVs and attaining a suppressed viral load in interrupting onward HIV transmission to susceptible partners - i.e. undetectable viral load = unable to transmit. From a population perspective, I think this is a vital point that is a consequence of the point you raise about people who are provided appropriate counselling and support, that they are more likely to disclose earlier and also more likely to adhere to treatment.

My main concern with this paper is the style in which it is written. The writing style, in my opinion, is not of an academic standard that would be acceptable to peer reviewed journals and needs to be tightened up considerably. There is a consistent use of 'non academic language' throughout the paper and whilst the content is sound, the presentation is not. e.g. 'What stood out for the most part' - this phrase is fine when speaking but does not read well in an academic paper. Another example, using words such a 'spread' in reference to HIV is not ideal, it would be preferable to use the word 'transmission'. I would encourage you to look closely at the expressions you have used and re-write the paper using acceptable academic phrasing (e.g. analysis was done - would read better as -analysis was undertaken). The grammar also needs some significant work.

There are some font issues - e.g. see lines 99 & 100

Also, line 297 - write vs as a word, i.e. versus

6. PLOS authors have the option to publish the peer review history of their article (what does this mean?). If published, this will include your full peer review and any attached files.

Reviewer #1: **Yes: **Fatch Welcome Kalembo

Reviewer #2: No

---

## [Author Response · Author response to Decision Letter 0]

13 May 2021

1. Line 59-61: Authors need to make it clear that disclosure of HIV status is one of the strategies that can help to prevent the transmission of HIV. The way it has been written, it is like disclosure of HIV status is the only way that can help to prevent the transmission of HIV

Response: Agreed and changed (Line 57-59)

2. The sentence covered by lines 65-67 needs citation.

Response: Agreed and added reference (Line 67)

3. It is important to describe the operational definition of HIV disclosure in this study as this can help to guide the interpretation of the findings

Response: Agreed and added in introduction (Line 60-61)

4. The justification for conducting this study is not convincing or correct. Authors identify lack of studies that have assessed barriers to timely disclosure. However, contrary to the authors’ claim there are several studies in Tanzania that have been conducted related to barriers to disclosure of HIV status to others or sexual partners some of them include the following (Damian et al., 2019; Hallberg et al., 2019; Maluka, 2014). Thus, authors need to be clear with the reasons for conducting this study. Why is it important to conduct this study? What will it add to the existing literature regarding disclosure of HIV status to others/sexual partners in Tanzania?

Response: Modified the wording on justification. 

There are studies that have shown the prevalence in the delay in disclosure, and some managed to get reasons but they were not explored in depth to get a deeper meaning qualitatively especially in Tanzania.

This study deals with the barriers to and facilitators of motivation for the timely disclosure. It explores the reasons why PLWHA have a delay in disclosing and what helped them eventually overcome it. 

Knowing the barriers that prevent timely disclosure will help counsellors address them to promote early disclosure to reap the benefits of timely disclosure such as ARV adherence, viral suppression and prevention of the spread of HIV.

The study by Hallberg et al and Damian et al were quantitative studies that focused on factors that affected disclosure, they did not study reason/barriers for timely disclosure. Study by Maluka too did not study timely disclosure (Line 98-117)

5. Where did data collection take place in the research site?

How were participants approached to take part in the interview?

How long did each interview take to be completed?

How many people were involved in the data analysis process? 

Response: Modified and added the information (Line 155-157

Line 152-153

Line 159-161

Line 190-192)

6. Given that no interview took place at Aga Khan Hospital, it is not necessary for the authors to include information about this facility in this article.

Response: I think it is important to include Aga Khan hospital. Lack of participants explains the social class theory that we mentioned in the discussion section. (Line 459 -473)

7. What type of questions were included in the interview guide. It would be important if authors can include the interview guide as supplementary material to their submission 

Response: Uploading the interview guide as supporting information – S1 and S2 (Line 159)

8. The findings in the three categories seem to be limited. For example, the last category ‘Consequences of delayed disclosure’ is too brief and lacks sufficient detail to be a standalone category. 

Response: Added details in the third category (Line 348 -364)

As a qualitative study, the findings of the study should have some depth to enable the readers to understand participants’ perspectives. The findings are also missing some important information that might be relevant to the study aim for example, what prompted participants to disclose their HIV status to the people they disclosed to?

Response: This was addressed in the sub-category motivation for disclose of serostatus where we discussed that participants would disclose to those they were close to and had established trust with them (Line 323 -330)

How did the significant others/family members reacted to the disclosure

Response: Included as negative and positive reactions/outcomes in the results section. 

Positive reactions were added under the subcategory motivations to disclose. This is because positive reactions encouraged further disclosure.

Negative reactions were placed under the subcategory Barriers hindering timely disclosure as these reactions prevented further disclosure 

(Line 331 -346, Line 256 - 288)

How was the disclosure done?

Response: Not part of my objectives

9. Authors have discussed some findings which are not presented in the findings section. For example, line 316-318, the authors have indicated that a majority of the participants experienced some sort of stigmatization (getting laughed at, discrimination); infidelity and one participant attempted suicide. This information is not described in the findings section.

Response: This is included in the findings under the subcategory barriers to disclose. Experiencing a negative outcome hindered and prevented further disclosure (Line 256-288)

10.The limitations of the study need more work. Aga Khan Hospital is not part of the research site in this study as none of the participants was recruited from this research site. Researchers should highlight the limitations of the research methodology used in this study.

Response: Our limitation was based on the fact that we got no participants from Aga khan. Which was why we discussed social class and perceived stigma – see comment 12 below. 

However I have added other limitations after discussion section

(Line 459-475)

11. The implications of findings for practice, research and policy are also not clearly explained in this section.

Response: Discussed in the conclusion

Implication of finding: strengthen the counselling services of HCW by training to promote timely disclosure.

Areas of research: stigma surrounding HIV, relationship of social class and HIV

( Line 493-497, Line 485-491)

12. Lines 352 to 256, authors have discussed social class in relation to stigma. However, social class has not been presented in the findings of the study. Authors need to discuss or make research recommendations based on the findings of the study

Response: Social class was brought up when we tried to understand why we were not able to recruit any participant from the Aga Khan hospital which caters for relatively higher class of patients. 

Our recommendation was to have other studies that see if there is a relation with HIV stigma and social class. Evidence show that social class in HIV does not receive much notice in literature and is a neglected area in research (see discussion)

(Line 464-473)

13. All sections of the paper need editing

Response: Thank you for your comments and review

14. Congratulations on an interesting study, the findings of which should be of interest to others working in Africa.

The methods you have used and have described in your paper are sound and appropriate.

The results, whilst not new, are important as they highlight the need for consistent high quality counselling of PLWHA in order to disclose early and interrupt onward transmission to sexual partners.

The points you raise in your discussion are clear and reference other appropriate and relevant research. I suggest you consider including in the discussion some thoughts about the importance of adherence to ARVs and attaining a suppressed viral load in interrupting onward HIV transmission to susceptible partners - i.e. undetectable viral load = unable to transmit. 

From a population perspective, I think this is a vital point that is a consequence of the point you raise about people who are provided appropriate counselling and support,that they are more likely to disclose earlier and also more likely to adhere to treatment.

My main concern with this paper is the style in which it is written. The writing style, in my opinion, is not of an academic standard that would be acceptable to peer reviewed journals and needs to be tightened up considerably. There is a consistent use of 'non academic language' throughout the paper and whilst the content is sound, the presentation is not. e.g. 'What stood out for the most part' - this phrase is fine when speaking but does not read well in an academic paper. Another example, using words such a 'spread' in reference to HIV is not ideal, it would be preferable to use the word 'transmission'. I would encourage you to look closely at the expressions you have used and re-write the paper using acceptable academic phrasing (e.g. analysis was done - would read better as -analysis was undertaken). The grammar also needs some significant work.

There are some font issues - e.g. see lines 99 & 100

Also, line 297 - write vs as a word, i.e. versus

Response: We have included the importance of ARV in the discussion

We also have edited the grammatical and vocabulary errors throughout the paper using academic language and tracked the changes 

Corrected the font

(Line 381 -385)

---

## [Decision Letter · Decision Letter 1]

1 Jun 2021

PONE-D-20-37341R1

Barriers to timely disclosure of HIV serostatus: A qualitative study at care and treatment centers in Dar es Salaam, Tanzania.

PLOS ONE

Dear Dr. Matillya,

Thank you for submitting your manuscript to PLOS ONE. After careful consideration, we feel that it has merit but does not fully meet PLOS ONE’s publication criteria as it currently stands. Therefore, we invite you to submit a revised version of the manuscript that addresses the points raised during the review process.

We look forward to receiving your revised manuscript.

Kind regards,

Joel Msafiri Francis, MD, MS, PhD

Academic Editor

PLOS ONE

Journal Requirements:

Reviewers' comments:

Reviewer's Responses to Questions

**Comments to the Author**

1. If the authors have adequately addressed your comments raised in a previous round of review and you feel that this manuscript is now acceptable for publication, you may indicate that here to bypass the “Comments to the Author” section, enter your conflict of interest statement in the “Confidential to Editor” section, and submit your "Accept" recommendation.

Reviewer #1: (No Response)

Reviewer #2: All comments have been addressed

2. Is the manuscript technically sound, and do the data support the conclusions?

Reviewer #1: Yes

Reviewer #2: Yes

3. Has the statistical analysis been performed appropriately and rigorously? 

Reviewer #1: N/A

Reviewer #2: N/A

4. Have the authors made all data underlying the findings in their manuscript fully available?

Reviewer #1: (No Response)

Reviewer #2: (No Response)

5. Is the manuscript presented in an intelligible fashion and written in standard English?

Reviewer #1: No

Reviewer #2: Yes

6. Review Comments to the Author

Reviewer #1: The authors have addressed issues identified in the previous review. However, The paper needs thorough editing to make it suitable for publication as there are many typos and grammar errors that are obscuring the clarity of the paper.

In the conclusion section, page 34 line 489-490 reads, “This paves the way to achieve two parameters of the 90-90-90 target set by UNAIDS; namely, easier access and linkage to medications and subsequent viral suppression.” Authors need to be aware that the 90-90-90 target set by UNAIDS finished in 2020. UNAIDS set another target, the 95:95:95 target to end the AIDS epidemic by 2030.

Reviewer #2: Thank you for addressing my previous comments.

One thing that surprises me is that none of the participants in your study questioned where they had acquired the HIV infection from - for example - none suspected they had acquired it from their sexual partner and therefore questioned that partner's fidelity. This is a common occurrence in other countries, especially from women who suspect their husband/partner has had sex with another (infected) person. Perhaps you could add a sentence or two about why the participants in your study didn't suspect their partner's of infecting them - did all participants have more than one sexual partner? You would also need to comment about whether or not your participants had had a previous negative HIV test and could therefore estimate the period of time in which they became infected. e.g addressing whether Tanzania routinely test antenatal women for HIV?

Otherwise, Congratulations on an interesting paper.

7. PLOS authors have the option to publish the peer review history of their article (what does this mean?). If published, this will include your full peer review and any attached files.

Reviewer #1: **Yes: **Fatch Kalembo

Reviewer #2: No

---

## [Author Response · Author response to Decision Letter 1]

30 Jun 2021

Reviewer: The authors have addressed issues identified in the previous review. However, The paper needs thorough editing to make it suitable for publication as there are many typos and grammar errors that are obscuring the clarity of the paper.

Response: Have resubmitted it to English editors to make it ready for publication

Reviewer: In the conclusion section, page 34 line 489-490 reads, “This paves the way to achieve two parameters of the 90-90-90 target set by UNAIDS; namely, easier access and linkage to medications and subsequent viral suppression.” Authors need to be aware that the 90-90-90 target set by UNAIDS finished in 2020. UNAIDS set another target, the 95:95:95 target to end the AIDS epidemic by 2030

Response: Thank you for pointing that out. We have rectified it both in the introduction and conclusion.

Reviewer #2: Thank you for addressing my previous comments.

One thing that surprises me is that none of the participants in your study questioned where they had acquired the HIV infection from - for example - none suspected they had acquired it from their sexual partner and therefore questioned that partner's fidelity. This is a common occurrence in other countries, especially from women who suspect their husband/partner has had sex with another (infected) person. Perhaps you could add a sentence or two about why the participants in your study didn't suspect their partner's of infecting them – 

Response: Thank you so much for insightful feedback and comments about our manuscript. This study main aim was to explore the barriers contributing to delayed disclosure and the reasons that made participants overcome it and finally disclose. Since it was not the focus of our study, we did not ask if they questioned where they had acquired the infection from and we do not have data for such. 

However, the data shows that, after disclosing to partners they were blamed to have been the ones who brought the disease into the relationship. This led to a strain and in some cases a break in the relationship. This is mentioned in the barriers of disclosure under the negative outcomes experienced.

Reviewer: did all participants have more than one sexual partner? 

Response: From the interviews we did, the participants all had one partner at time of disclosure.

Mentioned that in the results section 

Reviewer: You would also need to comment about whether or not your participants had had a previous negative HIV test and could therefore estimate the period of time in which they became infected. e.g addressing whether Tanzania routinely test antenatal women for HIV?

Otherwise, Congratulations on an interesting paper.

Response: All participants that were interviewed had not tested prior and so did not have a negative test before that. We have added a couple of lines in the discussion section about the introduction of HIV testing in routine antenatal and RCH services in Tanzania

We appreciate the feedback that has helped improve the quality of our manuscript.

---

## [Decision Letter · Decision Letter 2]

15 Jul 2021

PONE-D-20-37341R2

Barriers to timely disclosure of HIV serostatus: A qualitative study at care and treatment centers in Dar es Salaam, Tanzania.

PLOS ONE

Dear Dr. Matillya,

Thank you for submitting your manuscript to PLOS ONE. After careful consideration, we feel that it has merit but does not fully meet PLOS ONE’s publication criteria as it currently stands. Therefore, we invite you to submit a revised version of the manuscript that addresses the points raised during the review process.

Thank you for addressing the previous comments. Please kindly address a few minor comments in relation to proof reading and final editing the paper.

We look forward to receiving your revised manuscript.

Kind regards,

Joel Msafiri Francis, MD, MS, PhD

Academic Editor

PLOS ONE

Journal Requirements:

Reviewers' comments:

Reviewer's Responses to Questions

**Comments to the Author**

1. If the authors have adequately addressed your comments raised in a previous round of review and you feel that this manuscript is now acceptable for publication, you may indicate that here to bypass the “Comments to the Author” section, enter your conflict of interest statement in the “Confidential to Editor” section, and submit your "Accept" recommendation.

Reviewer #1: All comments have been addressed

Reviewer #2: All comments have been addressed

2. Is the manuscript technically sound, and do the data support the conclusions?

Reviewer #1: Yes

Reviewer #2: Yes

3. Has the statistical analysis been performed appropriately and rigorously? 

Reviewer #1: Yes

Reviewer #2: N/A

4. Have the authors made all data underlying the findings in their manuscript fully available?

Reviewer #1: (No Response)

Reviewer #2: No

5. Is the manuscript presented in an intelligible fashion and written in standard English?

Reviewer #1: No

Reviewer #2: Yes

6. Review Comments to the Author

Reviewer #1: The authors have addressed the comments I made in my previous review. Nonetheless, there are still several areas in the manuscript where there is no spacing between words or between words and references e.g. lines 52, 57, 51, 89, 172. Line 85 has a different font size from the rest of the manuscript.

Reviewer #2: (No Response)

7. PLOS authors have the option to publish the peer review history of their article (what does this mean?). If published, this will include your full peer review and any attached files.

Reviewer #1: **Yes: **Fatch W Kalembo

Reviewer #2: No

---

## [Author Response · Author response to Decision Letter 2]

7 Aug 2021

Reviewer #1: The authors have addressed the comments I made in my previous review. Nonetheless, there are still several areas in the manuscript where there is no spacing between words or between words and references e.g. lines 52, 57, 51, 89, 172. Line 85 has a different font size from the rest of the manuscript.

Response: Thank you for your observation. 

We have rectified all the spacing issues and fonts.

---

## [Editor Report · Decision Letter 3]

10 Aug 2021

Barriers to timely disclosure of HIV serostatus: A qualitative study at care and treatment centers in Dar es Salaam, Tanzania.

PONE-D-20-37341R3

Dear Dr. Matillya,

We’re pleased to inform you that your manuscript has been judged scientifically suitable for publication and will be formally accepted for publication once it meets all outstanding technical requirements.

Kind regards,

Joel Msafiri Francis, MD, MS, PhD

Academic Editor

PLOS ONE
---

## [Editor Report · Acceptance letter]

12 Aug 2021

PONE-D-20-37341R3 

Barriers to timely disclosure of HIV serostatus: A qualitative study at care and treatment centers in Dar es Salaam, Tanzania 

Dear Dr. Matillya:

I'm pleased to inform you that your manuscript has been deemed suitable for publication in PLOS ONE. Congratulations! Your manuscript is now with our production department. 

Kind regards, 

on behalf of

Dr. Joel Msafiri Francis 

Academic Editor

PLOS ONE